# Au@^109^Pd Core–Shell Nanoparticles Conjugated to Panitumumab for the Combined β^−^—Auger Electron Therapy of Triple-Negative Breast Cancer

**DOI:** 10.3390/ijms252413555

**Published:** 2024-12-18

**Authors:** Nasrin Abbasi Gharibkandi, Agnieszka Majkowska-Pilip, Rafał Walczak, Mateusz Wierzbicki, Aleksander Bilewicz

**Affiliations:** 1Centre of Radiochemistry and Nuclear Chemistry, Institute of Nuclear Chemistry and Technology, Dorodna 16 St., 03-195 Warsaw, Poland; n.abbasi@ichtj.waw.pl (N.A.G.); r.walczak@ichtj.waw.pl (R.W.); 2Department of Nuclear Medicine, Central Clinical Hospital of the Ministry of the Interior and Administration, Wołoska 137 St., 02-507 Warsaw, Poland; 3Institute of Biology, Warsaw University of Life Sciences, Ciszewskiego 8 St., 02-786 Warsaw, Poland; mateusz_wierzbicki@sggw.edu.pl

**Keywords:** ^109^Pd/^109m^Ag in vivo generator, radioimmunotherapy, triple-negative breast cancer

## Abstract

Apart from HER2-positive, triple-negative breast cancer (TNBC) is the second most highly invasive type of breast cancer. Although TNBC does not overexpress HER2 receptors, it has been observed that EGFR protein expression is present in this specific type of tumor, making it an attractive target for immune and radiopharmaceutical treatments. In our current study, we used ^109^Pd (T_1/2_ = 13.7 h) in the form of a ^109^Pd/^109m^Ag in vivo generator as a source of β^−^ particles and Auger electrons in targeted radionuclide therapy for TNBC. ^109^Pd, obtained through neutron irradiation of the ^108^Pd target, was deposited onto 15 nm gold nanoparticles to form Au@^109^Pd core–shell nanoparticles, which were then conjugated to the panitumumab antibody. Au@^109^Pd-PEG-panitumumab nanoparticles were bound, internalized, and partially routed to the nucleus in MDA-MB-231 human breast cancer cells overexpressing EGFR receptors. The Au@^109^Pd-panitumumab radioconjugate significantly reduced the metabolic activity of MDA-MB-231 cells in a dose-dependent manner. In conclusion, we have found that Au@^109^Pd-PEG-panitumumab nanoparticles show potential as a therapeutic agent for combined β^−^–Auger electron targeted radionuclide therapy of TNBC. The simultaneous emission of β^−^, conversion, and Auger electrons from the ^109^Pd/^109m^Ag generator, similar to ^161^Tb conjugates, significantly enhances the therapeutic effect. The partial localization of these nanoparticles into the cell nucleus, provided by the panitumumab vector, ensures effective therapy with Auger electrons. This is particularly important for the treatment of drug-resistant TNBC cells.

## 1. Introduction

Breast cancer is the most prevalent cancer in women, with 2,296,840 new cases and 670,000 fatalities worldwide recorded in 2022 [1]. The molecular classification of breast cancer relies on the overexpression pattern of estrogen, progesterone, and human epidermal growth factor receptor 2 (HER2). Among all types of breast cancer, 15–20% are HER2-positive, characterized by their highly aggressive and metastatic nature. This form of cancer progresses rapidly and frequently leads to distant metastases. However, due to the high concentration of the HER2 antigen on the cell surface, the cancer responds favorably to targeted therapy.

The second type of invasive breast cancer is known as triple-negative breast cancer (TNBC). This aggressive form of breast cancer is characterized by the absence of estrogen, progesterone receptors, and HER2 overexpression [2]. TNBC accounts for around 10–15% of all breast cancer cases, often showing rapid growth and a higher likelihood of having spread by the time of diagnosis, limited treatment options, and typically poor prognosis. Additionally, it has a greater tendency to recur after treatment compared to other forms of breast cancer. Due to the absence of specific receptors, hormone therapy and HER2-targeted drugs are not effective for women with triple-negative breast cancer. Therefore, cytotoxic chemotherapy is the only form of systemic therapy currently available for patients with TNBC. The FDA has approved anti-metabolites, paclitaxel, and anthracyclines as adjuvant and neoadjuvant chemotherapy regimens for patients with TNBC [3,4]. However, the toxicity of chemotherapy is harmful to patients, and many still do not receive clinical benefit. Therefore, there is a strong interest in identifying new targets for treating patients with TNBC [5]. 

Although TNBC does not overexpress HER2 receptors, it has been observed that EGFR protein expression is present in this specific type of tumor. EGFR, also known as HER1, is one of the four members of the HER family of receptors, along with HER2, HER3, and HER4. Further evidence supporting the significance of EGFR in TNBC comes from the discovery that high expression of EGFR predicts a poor prognosis in patients with this type of cancer. Like HER2, there are now multiple therapies available that target EGFR. These therapies include monoclonal antibodies such as cetuximab and panitumumab. They work by preventing ligand binding. Both of these antibodies have been approved for treating advanced colorectal cancers. Currently, intensive preclinical and phase I and II clinical trials are being conducted on their application in TNBC therapy.

Panitumumab, a fully humanized monoclonal antibody, exhibits a high affinity (K_d_ = 5 × 10^−11^ M) for the EGFR receptor. By blocking the binding of EGF ligands and TGF-α to EGFR, panitumumab inhibits tumor growth and triggers tumor regression [3]. Therapeutic properties of both panitumumab itself and its conjugates with radionuclides, in which it acts as a targeting vector to EGFR receptors, are under study.

In a series of studies, Reilly’s group at the University of Toronto has been investigating the use of panitumumab labeled with ^111^In, an Auger electron emitter, for targeted radionuclide therapy (TRT) of TNBC. The ^111^In radionuclide (with a half-life of 2.8 days) decays by electron capture, emitting 7.4 Auger electrons and 0.2 low-energy internal conversion (IC) electrons per decay which can be utilized for TRT. The Auger electron releases all of its energy over very short distances, leading to high linear energy transfer (LET: 1–23 keV/μm). This is effective in causing lethal DNA double-strand breaks in cancer cells, especially when the electrons are emitted close to DNA strands [6]. The AE can also be targeted at the cell membrane when radiolabeled antagonists of this receptor are used. The short range of AE and high LET make these electrons attractive for treatment of single tumor cells and micrometastases, preventing TNBC relapse. In one study [7], the use of panitumumab conjugated with the DOTA chelator and labeled with ^111^In resulted in high binding affinity to EGFR (K_D_ = 0.6 ± 0.2 × 10^−9^ M). This conjugate was found to be bound, internalized, and partially imported into the nucleus of human TNBC MDA-MB-231 and MDA-MB-468 cells. The emission of AE by ^111^In inside the nucleus of TNBC cells caused a significant increase in DNA DSBs, leading to cell cycle arrest in the G2/M phase and promoting apoptosis, which in turn reduced clonogenic survival in vitro. Additionally, in a mouse model with induced MDA-MB-231 cell tumors in the mammary fat pad, it significantly slowed tumor growth and prolonged survival compared with control mice. Subsequent studies [8] continued these investigations in the context of using ^111^In-DOTA-panitumumab in adjuvant therapy to improve survival in the relapse and metastatic progression treatment model.

Due to the limited number of chelator molecules that can be attached to a protein like panitumumab, the specific activity that can be achieved is limited. As a result, as few as one in 50 antibody molecules may be radiolabeled [9]. One strategy to overcome this delivery barrier may be to increase the number of chelators attached to the antibody molecules carrying a radionuclide, thereby increasing the amount of radioactivity targeted to tumor cells per receptor. To achieve this, panitumumab was attached to a chelating block polymer [10] or gold nanoparticles [11] carrying a large number of chelators. 

In several of our previous publications, we have enhanced the specific activity of similar antibody (trastuzumab) conjugates by utilizing radioactive metallic gold, platinum, or palladium nanoparticles, which are synthesized from radioactive ^198^Au [12,13], ^193/195m^Pt [14], or ^109/103^Pd [15] salts. In this way, we significantly increased the specific activity of trastuzumab radiobioconjugates, up to 40,000 radioactive ^198^Au atoms, when using 30 nm Au nanoparticles [13].

In our last published work in this field [15], we presented the outcomes of our studies exploring the application of a radiobioconjugate consisting of gold nanoparticles coated with a layer of ^109^Pd (Au@^109^Pd). This radiobioconjugate was conjugated with the monoclonal antibody trastuzumab and was applied in the targeted radionuclide therapy of HER2+ cancer. 

Palladium-109 exhibits significant potential for application in radionuclide therapy. It undergoes β^-^ decay (β_max_ = 1.12 MeV, 100% yield) to form ^109m^Ag, which has a half-life of 39.6 s. The resulting metastable ^109m^Ag decays to the stable isotope ^109^Ag, emitting photons at 88 keV (3.6%), followed by a cascade emission of both conversion and Auger electrons. The combination of ^109m^Ag with ^109^Pd forms an in vivo generator that emits medium-energy β^-^ particles along with a high quantity (18) of Auger/conversion electrons. The described properties of ^109^Pd closely resemble those of the widely used ^161^Tb (β^-^ and Auger electron emitter), which is currently the preferred option in targeted radiotherapy [16]. It allows for the use of ^161^Tb in both low- and high-linear energy transfer (LET) internal radiation therapy. Such treatment can simultaneously destroy large tumors using beta radiation and also allow for the destruction of small tumor metastases and subpopulations of treatment-resistant cancer stem cells using Auger electrons.

The results from our studies on the core–shell Au@^109^Pd nanoparticles–trastuzumab conjugate [15,17] showed that the ^109^Pd^/109m^Ag in vivo generator is more cytotoxic compared to Au@PdNPs labeled with either ^125^I (Auger emitter) or ^198^Au (β^−^ emitter). It is worth noting that despite the lack of nuclear localization, which is crucial for efficient Auger electron therapy [18], an adequate cytotoxic effect was achieved. These encouraging properties of the ^109^Pd@^109m^Ag-trastuzumab radiobioconjugate for treating HER2+ tumors prompted us to expand our studies to another aggressive form of breast cancer described above, TNBC. By substituting trastuzumab with panitumumab in the radioconjugate, we anticipated achieving improved cytotoxicity outcomes. Our optimism was based on the ability of panitumumab to partially localize (>10%) in the nucleus of MDA-MB-231 cells, potentially allowing it to be close to the DNA strand. It also extensively localizes in the cell membrane (70%), which is also a good target for Auger electron therapy [7]. This contrasts with the trastuzumab conjugate, which mainly localizes in the cytoplasm in the perinuclear region.

## 2. Results and Discussion 

### 2.1. Production of ^109^Pd Radionuclide

A large amount of ^109^Pd activity can be produced in a nuclear reactor through the ^108^Pd (n,γ)^109^Pd reaction, with a cross-section of 12.2 b. When natural palladium is used as the target material, more than 500 MBq/mg of ^109^Pd was obtained after 7 h of irradiation at a flux of 1.5 × 10^14^ n cm^−2^ s^−1^. Using a 98%-enriched ^108^Pd target, the obtained radioactivity exceeded 2 GBq/mg. When natural Pd targets are irradiated, a small amount of ^111^Ag (T_1/2_ = 7.45 d) is formed in the reaction ^110^Pd(n,γ)^111m^Pd→^111^Ag. The separation of ^111^Ag by co-precipitation with AgCl, as proposed by Das et al. [19], was highly effective. After the radiochemical processing, the gamma-ray spectrum did not show any characteristic gamma photo-peaks from ^111^Ag and other radioactive impurities. The target material was processed in about 40 min, resulting in only a 5% decrease in the initial activity. In the case of irradiating the isotopically enriched target, the negligible radioactivity of ^111^Ag made the removal procedure unnecessary.

### 2.2. Synthesis of Au@^109^PdNPs

To ensure that Auger electrons can travel a minimal distance (2–500 nm [20]), it is crucial for the emitter’s atoms to be located on the surface of the nanoparticle or in close proximity to it. Therefore, in our approach, to immobilize ^109^Pd atoms in nanoparticles, we selected 15 nm gold nanoparticles covered with one or a few palladium layers. The process of creating these core–shell nanoparticles involves the gradual, uniform deposition and rapid spread of reduced palladium atoms over the core region. Subsequently, the palladium atoms can migrate to the gold region and form a homogeneous layer, covering most of the AuNP surface [21]. The TEM images confirmed that the distribution of Pd was homogeneous and there were no locally elevated palladium clusters presented in our previous publication [15]. Due to the very short path length of Auger and conversion electrons, covering AuNPs with a thin layer of Pd was crucial to ensure their beneficial biological effects on tumor cells. Our calculations show that the obtained Au@Pd nanoparticles have an average of seven layers of Pd atoms on their surface. Considering the diameter of a Pd atom to be 0.274 nm, the total thickness of seven layers of Pd atoms should not exceed 2 nm. Furthermore, this should not significantly affect the energy of most Auger and conversion electrons emitted by ^109^Pd. By increasing the specific activity, it would be feasible to reduce the thickness to just a single layer.

### 2.3. Synthesis of Au@Pd-PEG-COOH and Au@Pd-PEG-Panitumumab Bioconjugate 

In order to generate a radiobioconjugate having high affinity to EGFR receptors, the Au@Pd nanoparticles have been modified by the monoclonal antibody panitumumab using a polyethylene glycol (PEG) linker. Panitumumab was conjugated with PEG (ortho-pyridyldisulfde-PEG-succinimidyl carboxymethyl ester, OPSS-PEG-NHS) through lysine and next was attached to the Au@PdNPs by the formation of strong gold–thiol bonds. At last, to improve the dispersity of the bioconjugate, PEGylation was carried out with HS-PEG-COOH (5 kDa). The scheme of the developed Au@^109^Pd-PEG-panitumumab radiobioconjugate is shown in Figure 1. The increase in hydrodynamic diameter, along with the addition of PEG and panitumumab, demonstrates the effective conjugation of biomolecules, as shown in Table 1. Also, the difference in zeta potential values between citrate-coated AuNPs and AuNP-HS-PEG-panitumumab confirms the effectiveness of the surface modification. The radiochemical purity of the Au@^109^PdNP-PEG-panitumumab radiobioconjugate, after centrifugation from the reaction mixture, exceeds 99%.

In order to determine the mean number of attached panitumumab molecules to a single Au@PdNP, ^131^I-labeled panitumumab molecules were conjugated to Au@PdNPs [22]. Using this method, we calculated that an average of 19 panitumumab molecules were conjugated to a single nanoparticle. This calculation was performed assuming a spherical shape for the nanoparticle with an average diameter of 15 nm, as determined by TEM, with a gold density of 19.28 g/cm^3^. The data obtained from the DLS analysis indicate that the distribution is random.

### 2.4. Colloidal, Chemical, and Radiochemical Stability of Au@Pd-PEG-Panitumumab Conjugate

As the stability of synthesized radiobioconjugates is crucial for their potential use as radiopharmaceuticals, we conducted an assessment of their tendency to agglomerate in biological fluids. The zeta potential of Au@PdNP-PEG-panitumumab of −25.0 ± 1.8 mV should indicate a lack of stability of the bioconjugate, as ζ-potential values > ±30 mV are considered to indicate colloidal stability. However, our observation of the hydrodynamic diameter of Au@PdNP-PEG-panitumumab over 16 days (Figure 2) indicates no agglomeration or disintegration of the bioconjugates within the evaluation interval, indicating their high stability. This indicates that panitumumab and PEG molecules do not dissociate from the nanoparticle.

The radiochemical stability of Au@^109^PdNPs was tested using γ-spectrometry. After centrifugation, we did not observe the release of ^109^Pd and ^109m^Ag from the nanoparticles. The potential release of the daughter radionuclide due to nuclear decay is crucial for the use of in vivo generators in nuclear medicine. Our previous studies [17] show that, unlike chelator-based in vivo ^109^Pd/^109m^Ag generators, we observed complete retention of ^109m^Ag on Pd nanoparticles. This means that the parent radionuclide is integrated into a metallic nanoparticle rather than a chelate complex, and we did not detect any traces of ^109m^Ag released in water, PBS buffer (1 mM), or human serum.

### 2.5. EGFR Immunoreactivity

Increasing concentrations (0 to 20 nM) of Au@^109^PdNP-PEG-panitumumab were incubated with EGFR-overexpressing MDA-MB-231 cells, either in the presence or absence of a 500-fold molar excess of panitumumab. Figure 3 shows significant differences in binding with or without free panitumumab used to block the receptors, indicating specific EGFR-dependent binding to MDA-MB-231 cells; however, the SB (specific binding) curve did not reach saturation or even come close to saturation over the entire range of concentrations tested. The shape of the dependency curve shows notable distinctions compared to those of labeled antibodies or peptides, but it closely resembles the curves for nanoparticle conjugates with trastuzumab [14,23]. The characteristics of these curves are straightness across a broad concentration range and very large nonspecific binding that exceeds the specific binding. The phenomenon can be explained by the significant adhesion of nanoparticles on cell membranes, as demonstrated in multiple studies [24]. In contrast, the nonspecific binding of the tested monoclonal antibodies is typically minimal.

### 2.6. Internalization by Radiometric Assay and Confocal Imaging

In order to achieve optimal treatment results, it is crucial to deliver Auger-electron-emitting radionuclides to the cell nucleus, preferably close to the DNA. It has also been found that localization of the Auger electron emitter in the cell membrane can also cause cell death [6]. Nanoparticles carrying Auger emitters can be transported into the cell nucleus in two ways. The first way is through passive transport via nuclear pore complexes. These complexes allow the diffusion of ions, small molecules, and nanoparticles through aqueous channels with a diameter of approximately 9 nm. The main condition for transportation is that the substances must be hydrophilic. The second route is known as active transport. During this process, certain macromolecules are actively transported across the nuclear envelope. This transport is a highly selective process that can be divided into two steps: receptor binding, followed by translocation across the nuclear envelope [25]. Receptor binding is facilitated by nuclear localization signals, which have been identified in many nuclear proteins. 

In our previous studies [15], as well as in the studies conducted by Cai et al. [23], it was observed that trastuzumab and trastuzumab-modified Au@Pd and Au nanoparticles exhibited very high internalization (>90%) into SKOV-3, SKBR-3, and MDA-361 cells overexpressed the HER2 receptors. However, the localization of these conjugates into the cell nucleus was not observed. In the case of the behavior of Au@Pd-PEG-panitumumab NPs in the MDA-MB-231 cell overexpressing the EGFR receptor, the situation is different. Results of radiometric studies presented in Figure 4 indicated similar internalization into the cell; however, there is also a significant transport into the cell nucleus, in contrast to trastuzumab-modified nanoparticles in HER2+ cells, where transportation to the nucleus does not occur.

These results are well supported by confocal and dark-field microscopy images (Figure 5). The dark spots visible on the bright background represent agglomerated Au@Pd particles, whereas the green fluorescence signals indicate secondary mAb conjugated to panitumumab as an integral part of the bioconjugate. In dark-field microscopy, agglomerated Au@Pd-PEG-panitumumab NPs were observed in EGFR-targeted MDA-MB-231 cells, but not in untargeted Au@Pd-PEG. The possibility of observing Au@Pd-PEG-panitumumab NPs in dark-field mode in the absence of a signal for Au@Pd-PEG may be due to the clustering of receptor-bound Au@Pd-PEG-panitumumab NPs. This clustering could occur because some receptors have been found to cluster on lipid rafts on the cell surface [23]. In confocal microscopy images, we can also observe the accumulation of Au@Pd-PEG-panitumumab NPs inside cells, revealing the crucial role of panitumumab as a targeting vector for EGFR-overexpressed breast cancer cells. In contrast to the confocal microscopy images recorded for Au@Pd-PEG-trastuzumab in SKOV-3 cells, which displayed accumulation in the perinuclear area nearly to the nuclear envelope [15], we now observe a localization of the Au@Pd-PEG-panitumumab conjugate inside the cell nucleus. Therefore, the accumulation of NPs in close proximity to the most sensitive cellular organelle led us to expect high cytotoxicity induced by radioactive bioconjugates.

### 2.7. Cytotoxicity 

The viability of MDA-MB-231 cells overexpressing EGFR and incubated with non-radioactive Au@Pd-PEG-panitumumab NPs and Au@^109^Pd-PEG-panitumumab NPs was evaluated using the MTS assay. The bioconjugates of non-radioactive Au@Pd-PEG-panitumumab NPs were used at concentrations ranging from 11.25 μg/mL (3 × 10^−3^ nmol/mL NP) to 180 μg/mL (5 × 10^−2^ nmol/mL NP). The goal of this study was to investigate whether this bioconjugate could cause mitochondrial dysfunction and cell death. As shown in Figure 6a, the effect of non-radioactive Au@Pd-PEG-panitumumab NPs on the viability of MDA-MB-231 cells was observed across all tested concentrations. However, the most significant effect was observed for concentrations above 22.5 μg/mL. A similar cytotoxic effect of Au-PEG-panitumumab NP conjugates on TNBC was also observed by Yook et al. [11]. The authors suggested that the cytotoxicity of panitumumab nanoparticle conjugates on MDA-MB-231 cells may be due to the immunotoxic effect of the panitumumab antibody linked to the nanoparticles, as indicated by several clinical studies. They also proposed that the internalization of Au-PEG-panitumumab nanoparticles into the cell nucleus could potentially cause nuclear damage in cells, similar to what has been observed with AuNPs [26]. The cytotoxicity of PdNPs is significantly higher than that of AuNPs. Therefore, Au@Pd-PEG-panitumumab demonstrates increased cytotoxicity. This is consistent with our previous studies on the Au@Pd-PEG-trastuzumab conjugates on SKOV-3 cells [15], where nuclear localization of the conjugate in cell nuclei was not observed, and therefore, no nuclear damage caused by nanoparticles in cells was detected.

The effects of different levels of radioactivity concentrations of Au@^109^Pd-PEG-panitumumab and Au@^109^Pd-PEG conjugates on the metabolic viability of MDA-MB-231 cells at 24, 48, and 72 h after exposure are illustrated in Figure 6b,c. It is evident that the use of the Au@^109^Pd-panitumumab radiobioconjugate significantly reduced the metabolic activity of MDA-MB-231 cells, with the reduction depending on the dosage and time interval after administration. Increasing the radioactivity of the radiobioconjugate caused a gradual reduction in mitochondrial activity, leading to almost complete cell death for an activity of 40 MBq/mL. When analyzing the results shown in Figure 6b, it is important to consider that the individual solutions of Au@^109^Pd-panitumumab used in cytotoxic tests were obtained by diluting the initial solution of 40 MBq/mL, which contained 180 µg/mL of the conjugate. As a result, the observed total cytotoxic effect comprises a chemotoxic effect caused by the non-radioactive conjugate (Figure 6a) and a radiotoxic effect caused by β^-^ particles and Auger electrons emitted by ^109^Pd. 

In the case of Au@^109^PdNPs, despite the lack of panitumumab, a significant reduction in cell viability was found, particularly 72 h after exposure (Figure 6c). The range of β^-^ particles emitted by ^109^Pd in the body is approximately 4 mm. Therefore, the cross-fire effect of these particles could potentially target TNBC cells in tumors with low or moderate EGFR expression, or cells that are not effectively targeted by Au@^109^Pd-PEG-panitumumab NPs due to their uneven distribution within the tumor. However, this effect is much smaller compared to the vector-attached radiobioconjugates, where due to internalization, the chemotoxicity of panitumumab, Au@Pd nanoparticles, and the radiotoxicity of Auger electrons are additionally involved.

### 2.8. DNA Double-Strand Breaks (DSBs) 

Lethal and irreparable damage to genetic material is considered one of the most desired outcomes of radiopharmaceutical anticancer activity [27]. These damages can occur through direct ionization of DNA caused by ionizing radiation or by the interaction of reactive oxygen species (ROS) generated from water with the DNA strand. In our study, we investigated the formation of DSBs following exposure to Au@^109^Pd-PEG-panitumumab nanoparticles. These nanoparticles can induce DSBs through direct and indirect interactions of β^-^ and Auger electron radiation, as well as through the generation of ROS by Pd nanoparticles [28]. The presence of DSBs in MDA-MB-231 cells was confirmed by the detection of γH2A.X foci, offering a fast and sensitive method for detecting DSBs induced by various cytotoxic agents, including ionizing radiation. DNA double-strand breaks in MDA-MB-231 cells are visualized with γH2A.X foci after treatment with Au@^109^Pd-PEG-panitumumab NPs (Figure 7). Quantification of these foci, represented as the integrated density of γH2AX foci per nucleus area, is shown in Figure 8.

The presence of DSBs in the MA-MB-231 cells treated with Au@^109^Pd-PEG-panitumumab NPs is consistent with the cytotoxicity results shown in Figure 6b. Similarly, as observed in HepG2 cells treated with ^109^Pd-PEG NPs [17], the high-LET Auger and conversion electrons emitted by the ^109^Pd/^109m^Ag in vivo generator lead to the formation of a high number of DSBs. However, when comparing our results with those of ^109^Pd-PEG NPs on HepG2 cells, a significantly higher number of γH2A.X foci is observed for the Au@^109^Pd-PEG-panitumumab radiobioconjugate. For instance, at an activity of 20 MBq/mL, the average number of γH2A.X foci in the cell nucleus reaches eight for Au@^109^Pd-PEG-panitumumab; however, it was only three for ^109^Pd-PEG NPs on HepG2 cells. This effect is likely due to the additional immunotoxicity effect on MDA-MB-231 cells generated by the panitumumab antibody conjugated to the nanoparticles. In the studies of DSBs caused by the ^111^In-labeled conjugate of trastuzumab with dendrimers derivatized with multiple DTPA chelators [29], a similar number of γH2A.X foci was observed on cells overexpressing HER2 receptors (SK-Br-3 cells). There were six γH2A.X foci in the cell nucleus at 12 MBq/mL activity, while our study showed seven γH2A.X foci at 10 MBq/mL Au@^109^Pd-PEG-panitumumab on MDA-MB-231 cells. As ^111^In does not emit beta particles, in contrast to the ^109^Pd/^109m^Ag in vivo generator, it can be deduced that the presence of DSBs is associated with Auger electron radiation.

## 3. Materials and Methods

### 3.1. Chemical Reagents 

Gold (III) chloride trihydrate (HAuCl_4_·3H_2_O), trisodium citrate dihydrate (C_6_H_9_Na_3_O_9_), and HS-PEG-COOH (poly(ethylene glycol), 5 kDa) were purchased from Sigma-Aldrich (St. Louis, MO, USA), and OPSS-PEG-NHS (alpha-pyridyl2-disulfd-omega-carboxy succinimidyl ester poly(ethylene glycol), 5 kDa) was obtained from Creative PEGworks (Chapel Hill, NC, USA). Panitumumab (Vectibix^®^) from Amgen (Thousand Oaks, CA, USA) and a PD-10 column (GE Healthcare, Piscataway, NJ, USA) were also utilized in such studies. Hydrochloric acid and sodium hydroxide were purchased from POCH (Gliwice, Poland). Fluorescence mounting medium was obtained from Dako (Carpinteria, CA, USA). The following materials were used in cell studies: phosphate-buffered saline (PBS) and dimethylsulfoxide (DMSO) from Sigma-Aldrich (St. Louis, MO, USA), and the CellTiter 96^®^ Aqueous One Solution Reagent (MTS compound) from Promega (Mannheim, Germany). MDA-MB-231 cells were purchased from the American Type Tissue Culture Collection (ATCC, Rockville, MD, USA), cultured in DMEM medium supplemented with 10% FBS and 1% penicillin−streptomycin (all from Beth Haemek, Israel), and maintained at 37 °C and 5% CO_2_. Furthermore, the solutions were generated using double-distilled water (18.2 MΩ·cm, Hydrolab, Straszyn, Poland).

### 3.2. Radionuclides

^109^Pd was obtained by thermal neutron (1–2 × 10^14^ n/cm^2^ s) irradiation of a natural palladium target (~2 mg, metal powder) or ~1 mg, metal powder enriched to >99% ^108^Pd in the Maria nuclear reactor (Otwock-Świerk, Poland) for 7 h. After a cooling time of 12 h, the radioactive palladium target was dissolved in aqua regia (HNO_3_: HCl—1:3, 200–400 µL) and heated at 130 °C until near-evaporation. 

The elimination of the remaining nitrate was accomplished by using three 200 µL portions of 0.1 M HCl and double-distilled H_2_O, respectively, and then heating at 130 °C until near-evaporation. At last, the target was suspended in 0.1 M HCl and 0.5 M NaOH in a ratio of 7:1, to generate Na_2_PdCl_4_. Neutron irradiation of the natural Pd target can lead to the formation of ^111^Ag as an impurity in the reaction of ^110^Pd (n,γ)^111m^Pd→^111^Ag. It is essential to remove it from the solution before use, which was achieved by precipitating ^111^Ag as AgCl using AgNO_3_ in the modified procedure described by Das et al. [17,19].

^131^I radionuclide (no carrier added) was utilized for radioiodination of panitumumab, which was used to determine the number of panitumumab molecules attached to the Au@Pd surface. Na^131^I (with a specific activity of about >550 GBq/mg) was obtained from POLATOM Radioisotope Centre in Świerk, Poland.

### 3.3. Synthesis of Core–Shell Au@Pd and Radioactive Au@^109^PdNPs

Core–shell Au@Pd nanoparticles (NPs) were synthesized using the same procedure as described in [15]. Initially, gold nanoparticles with a diameter of 15 nm were synthesized based on the Turkevich method, with slight modifications by our group. In the next step, AuNPs were coated with an ultrathin layer of palladium via the chemical reduction of Na_2_PdCl_4_ by ascorbic acid using the procedure described before for the synthesis of Au@Pt [20]. Briefly, AuNPs solution was heated to 90 °C for 10 min. Subsequently, Na_2_PdCl_4_ (0.27 mg, 1 mM) or Na_2_^109^PdCl_4_ were added as the palladium precursors, along with ascorbic acid (7.5 mg, 2 M) as the reducing agent, at 10 min and 30 min intervals. The reaction mixture was then heated for 30 min at 90 °C, cooled to room temperature, and characterized using DLS and UV–Vis techniques.

### 3.4. Synthesis of Non-Radioactive and Radioactive Au@^109^Pd-PEG-Panitumumab Radiobioconjugate

The radioactive antibody-conjugated nanoparticles Au@^109^Pd-PEG-panitumumab were synthesized using the same procedure as for Au@^109^Pd-PEG-trastuzumab [15]. Panitumumab (1 mg) was combined with a 25-fold molar excess of OPSS-PEG-NHS (5 kDa) in carbonate buffer (100 mM) and allowed to react overnight. Afterward, the purified OPSS-PEG-panitumumab was conjugated to Au@Pd or Au@^109^Pd in carbonate buffer (20 mM) for 45 min. A 15000-molar excess of HS-PEG-COOH (5 kDa) was added to increase bioconjugate dispersity for 30 min at room temperature. At last, the product was purified by centrifugation, dispersed in deionized water, and characterized.

### 3.5. Determination of Number of Panitumumab Molecules Conjugated to Au@Pd Nanoparticles

We used iodinated panitumumab to calculate protein molecules conjugated to nanoparticles ([^131^I]I-panitumumab). The ^131^I labeling was performed using the Iodogen method [22]. In brief, 2 mg of panitumumab, 64 MBq of ^131^I, and 0.1 M PBS buffer were mixed with Iodogen reagent for 10 min in room temperature. The mixture was purified using a PD-10 column (Sephadex G-25 resin) and PBS buffer (10 mM) as a mobile phase. Consequently, buffer exchange was carried out using Vivaspin^®^500 centrifugal concentrators and the product was then transferred into an aqueous solution. Afterwards, 250 μg of [^131^I]I-panitumumab was mixed with OPSS-PEG-NHS (5 kDa, 25-fold molar excess) in carbonate buffer (100 mM) overnight, purified, and conjugated to nanoparticles. To determine the yield of conjugation, after centrifuging of nanoparticles, the obtained supernatant was collected, and the activity of both—nanoparticles and supernatant—portions was measured. Finally, the number of iodinated panitumumab molecules attached to one Au@PdNP was calculated by dividing the moles of protein by that of nanoparticles.

### 3.6. Stability Studies of Au@Pd-PEG-Panitumumab Bioconjugate Colloid 

The stability of the bioconjugate in physiological conditions was evaluated at 37 °C over a period of 16 days following centrifugation and dispersion of the Au@Pd-PEG-panitumumab in both PBS buffer and saline (0.9% NaCl). The aggregation tendency was investigated by measuring the hydrodynamic diameter and zeta potential using the DLS technique. The radiochemical stability was evaluated using γ spectrometry to detect any detachment of either ^109^Pd or ^109m^Ag from the core surface. For this purpose, radiobioconjugates were centrifuged after incubation in PBS, 0.9% NaCl, and human serum (HS), and the obtained fractions (supernatant and nanoparticles) were then quantified using gamma spectrometry. 

### 3.7. Determination of EGFR Immunoreactivity

The immunoreactivity of Au@^109^Pd-PEG-panitumubab with EGFR-overexpressing MDA-MB-231 cells was determined in a direct (saturation) binding assay. MBA-MB-231 cells (7.5 × 10^3^ cells/well) were seeded in 24-well plates and incubated overnight. The following day, plates were washed twice with cold PBS and incubated with increasing concentrations of Au@^109^Pd-PEG-panitumubab (0−20 nM) at 4 °C for 2 h. To determine non-specific binding, one of the plates was incubated with an excess of 500-fold molar of unlabeled mAb. After incubation, cells were washed twice with cold PBS and collected in tubes. The cells were then lysed twice with 800 μL of 1 M NaOH and collected in separate tubes. The samples were counted using the Wizard2 system.

### 3.8. Internalization Studies 

Internalization studies were performed on MDA-MB-231 cells. Briefly, the day before the experiment, 6× 10^5^ cells/well were seeded into 6-well plates. After 24 h of incubation, cells were rinsed with PBS and then treated with tested compounds (1 mL). To avoid internalization, the cells were incubated at 4 °C for 1 h. After that, the medium was collected as the unbound portion, and then the cells were washed with PBS and a fresh medium (1 mL) was added. Plates were then incubated (37 °C, 5% CO_2_) for various time points of interest, including 6, 18, and 24 h. To evaluate the membrane-bound portion, cells were rinsed twice with glycine-HCl buffer (pH ~2.8, 0.05 M) and then kept at 4 °C for 5 min. Finally, the internalized portion was collected by lysing the cells with 1 M NaOH. Non-specific binding was assessed with the same procedure as implemented for receptor binding affinity.

### 3.9. Confocal Microscopy Imaging 

MDA-MB-231 cells (2 × 10^5^ cells per well) were cultured on sterile glass coverslips with a diameter of 12 mm (Thermo Fischer Scientific, Waltham, MA, USA) in 6-well plates and then left to incubate over 24 h. The following day, after removal of the medium, cells were administered with panitumumab (73 µg/mL), Au@Pd-PEG-COOH (1.62 × 10^12^ NPs/mL), and Au@Pd-PEG-panitumumab bioconjugate (1.62 × 10^12^ NPs/mL), and incubated overnight. The cells were stained with both Hoechst 33,258 (λex/em = 352/454 nm) and an anti-human IgG secondary antibody coupled to FITC (λex/em = 490/525 nm), and bright-field images for nanoparticles were obtained by a transmitted light detector (T-PMT). All analysis was performed using Image J 1.54i software.

### 3.10. Cytotoxicity Studies 

Cytotoxicity experiments were conducted on MDA-MB-231 cells using the MTS assay. Briefly, cells were seeded at a density of 3 × 10^3^ cells per well in 96-well plates and incubated overnight at 37 °C with 5% CO_2_ atmosphere. The next day, after aspirating the medium, the cells were washed with PBS. Subsequently, antibody-conjugated nanoparticles at the concentrations of (11–180 µg/mL, for non-radioactive) and (180 µg/mL for 40 MBq/mL, 90 µg/mL for 20 MBq/mL, 45 µg/mL for 10 MBq/mL, for radioactive compounds) were suspended in fully supplemented growing medium and 100 µL per well was added and incubated for 24–72 h at 37 °C with 5% CO_2_ atmosphere. Before adding the MTS reagent, the medium was aspirated, the cells were washed with PBS, and fresh medium was added to the wells. Finally, the percentage of metabolically active cells was determined by the addition of CellTiter96^®^ AQueous One Solution Reagent and measurement of the absorbance at 490 nm.

### 3.11. DNA Double-Strand Break Study

γ-H2AX nuclear staining was performed to evaluate the intensity of DNA damage (double-strand breaks) induced by treatment with Au@^109^PdNPs-PEG-panitumumab. MDA-MB-231 cells (2 × 10^5^/well) were cultured in 6-well plates. Each well is covered with five sterile glass coverslips with a diameter of 12 mm. After 24 h of incubation, the medium was removed, and the cells were exposed to various concentrations (ranging from 0 to 40 MBq/mL) and staurosporine (0.5 μM) as a positive control. The cells were then incubated for 4 and 24 h. To identify γH2A.X foci 350 μL of a primary antiphospho-histone H2A.X (Ser139) antibody, clone JBW301 (with a dilution ratio of 1:100) was added to each well and left overnight at 4 °C. On the following day, the primary antibody was replaced with a secondary antibody (anti-mouse IgG), which was dissolved in blocking buffer (BB- 4% BSA in TBS) and labeled with CF™ 633. Subsequently, the cells were incubated at room temperature for 2 h. Afterward, the cells were washed with water and then stained with Hoechst 33258. The imaging study was carried out using an FV-1000 confocal microscope (Olympus Corporation, Tokyo, Japan) with ex/em maxima of 630/650 nm for CF633 and ex/em maxima of 352/454 nm for Hoechst 33258. The results were analyzed using the Fiji 2.9.0 version. 

### 3.12. Statistical Analysis 

Statistical analysis, one-way ANOVA, and Student’s t-tests were carried out with GraphPad Prism v.8 (GraphPad Software, San Diego, CA, USA). The results are expressed as mean ± SD. *p*-values are presented as follows: (*) *p* ≤ 0.05, (**) *p* ≤ 0.001, (***), and *p* ≤ 0.0001.

## 4. Conclusions 

In this article, we discuss the potential application of ^109^Pd-PEG-panitumumab nanoparticle conjugates as an in vivo ^109^Pd/^109m^Ag generator for targeted therapy of TNBC. Similar to previous studies, we show that, unlike most chelator-based generators, we observed complete retention of the daughter isotope on the nanoparticles. This has important implications for targeted therapy, as it does not result in the displacement of radioactivity outside the targeting area. The Au@^109^Pd-PEG-panitumumab nanoparticle conjugates have been found to exhibit multiple toxic interactions with cancer cells. They are both radiotoxic, emitting β^-^ and Auger electrons, and exhibit immunotoxicity due to the presence of the panitumumab antibody. Additionally, they demonstrate chemotoxicity as a result of the catalytic generation of ROS on PdNPs. However, the radiotoxic effects of Auger electrons become dominant due to the relatively high internalization of the radiobioconjugate into the cell nucleus, which is caused by the presence of the panitumumab vector. The emission of this high-LET radiation is particularly important for the treatment of resistant triple-negative breast cancer cells.

It is important to note that using radioactive bioconjugates based on metallic nanoparticles poses challenges for systemic administration. The accumulation of radioactively labeled nanobioconjugates in the liver, lungs, and spleen can expose these organs to radiation and reduce the accumulation of these radiobioconjugates by tumors. As a result, intravenous administration of Au@^109^Pd-PEG-panitumumab is not feasible. Many studies suggest that for these types of radiobioconjugates, localized administration directly to the tumor or to the resection cavity is more effective [9,11,13,15]. Despite these limitations, the concept of using an in vivo ^109^Pd/^109m^Ag generator and the promising results obtained encourage further exploration of this treatment strategy.

## Figures and Tables

**Figure 1 ijms-25-13555-f001:**
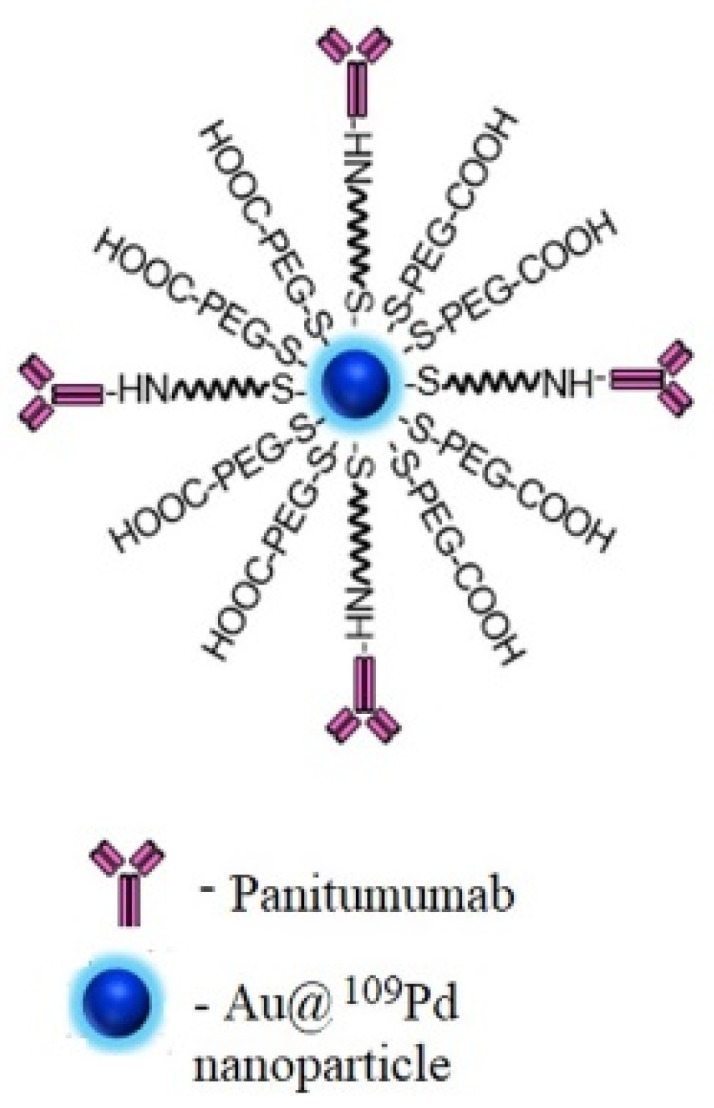
The scheme of the developed Au@^109^Pd-PEG-panitumumab radiobioconjugate.

**Figure 2 ijms-25-13555-f002:**
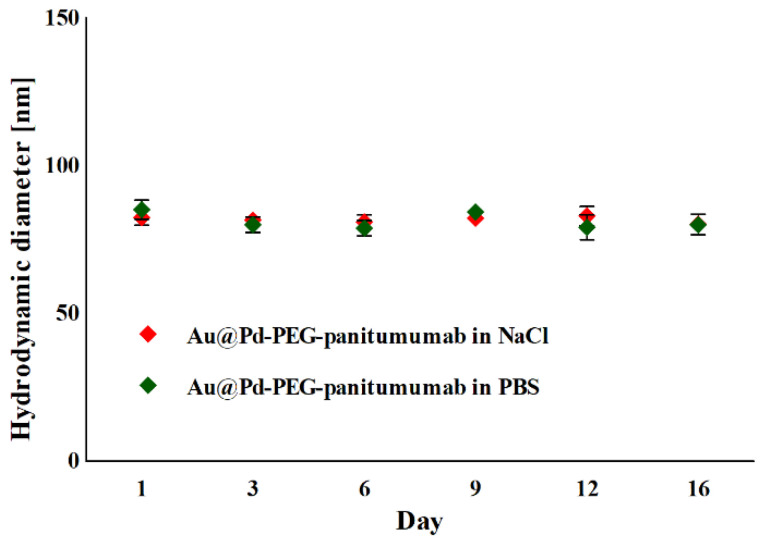
Changes in the hydrodynamic diameter of the Au@Pd-PEG-panitumumab nanoparticles incubated in 10 mM PBS buffer and 0.9% NaCl.

**Figure 3 ijms-25-13555-f003:**
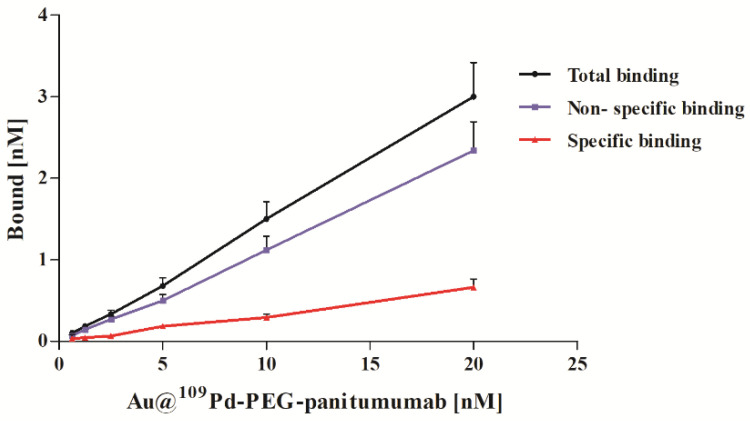
Binding studies of Au@^109^PdNP-PEG-panitumumab on MDA-MB-231 cell line.

**Figure 4 ijms-25-13555-f004:**
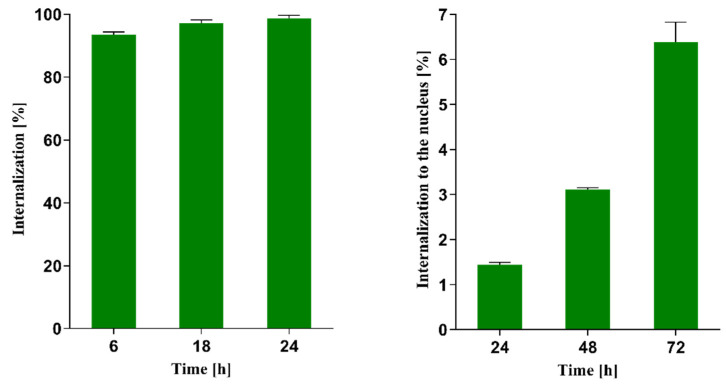
Internalization of Au@Pd-PEG-panitumumab NPs into the MDA-MB-231 cell overexpressing EGFR receptor (**left**) and intranuclear uptake of radioconjugate (**right**).

**Figure 5 ijms-25-13555-f005:**
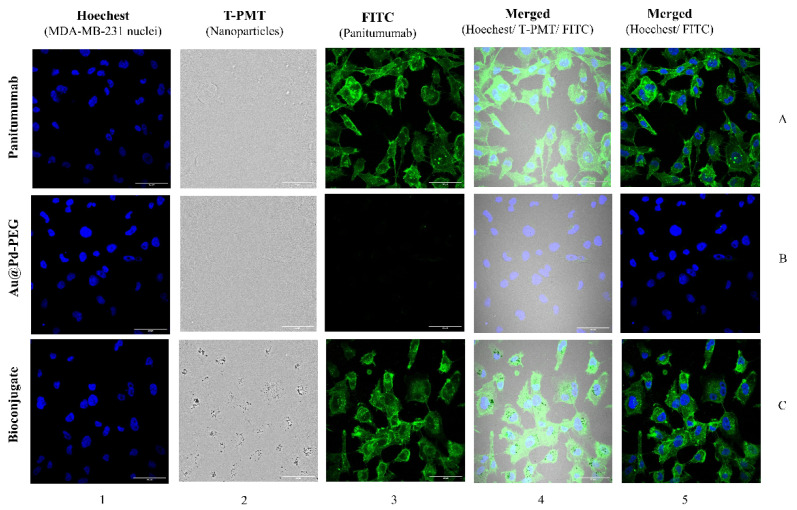
Internalization of panitumumab, Au@Pd-PEG-COOH, and Au@Pd-PEG-panitumumab in MDA-MB-231 cells determined by confocal microscopy. The fluorescence signals indicate the following: subcellular panitumumab distribution (green) and nuclei intracellular localization (blue). Au@Pd-containing particles (dark spots) were also visualized with a transmitted light detector (T-PMT).

**Figure 6 ijms-25-13555-f006:**
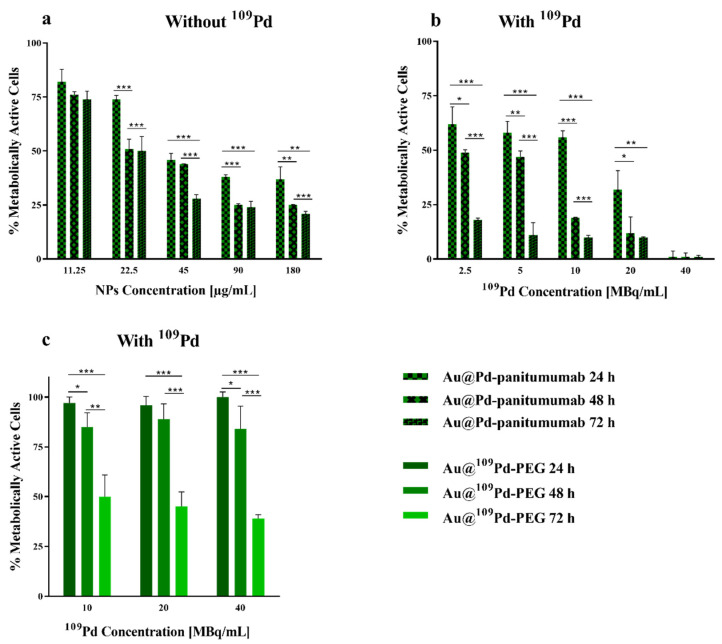
Metabolic viability of MDA-MB-231 cells after treatment with different concentrations of Au@Pd-PEG-panitumumab non-radioactive conjugates (**a**) and with different radioactive doses of Au@^109^Pd-PEG-panitumumab NPs (**b**) and of Au@^109^Pd-PEG radioactive conjugates (**c**) after 24 h, 48 h, and 72 h of incubation. ). The results are expressed as mean ± SD. *p*-values are presented as follows: (*) *p* ≤ 0.05, (**) *p* ≤ 0.001, (***), and *p* ≤ 0.0001.

**Figure 7 ijms-25-13555-f007:**
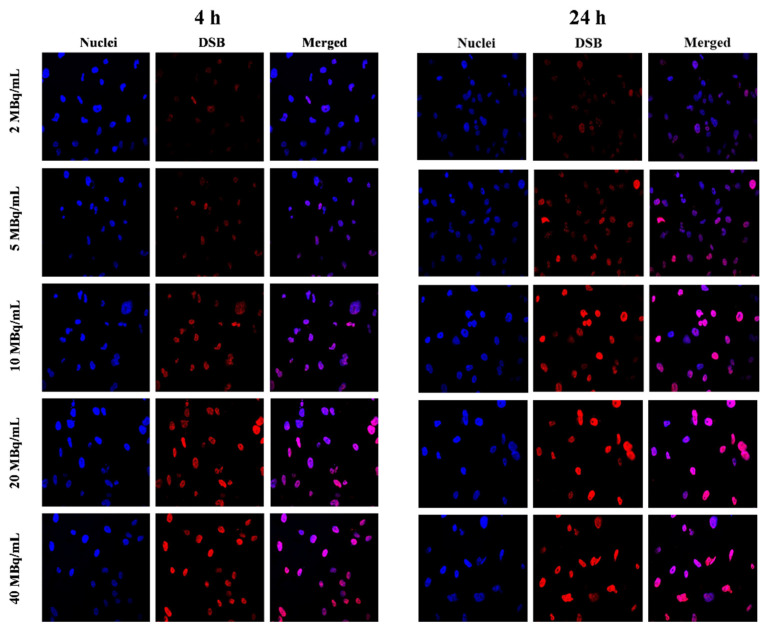
Confocal immunofluorescence microscopy exhibiting γH2A.X foci (red) in the nucleus (counterstained blue with DAPI) and merged (purple) of MDA-MB-231 breast cancer cells treated with various radioactivity of Au@^109^Pd-PEG-panitumumab recorded after 4 h (**left**) and 24 h (**right**) of incubation.

**Figure 8 ijms-25-13555-f008:**
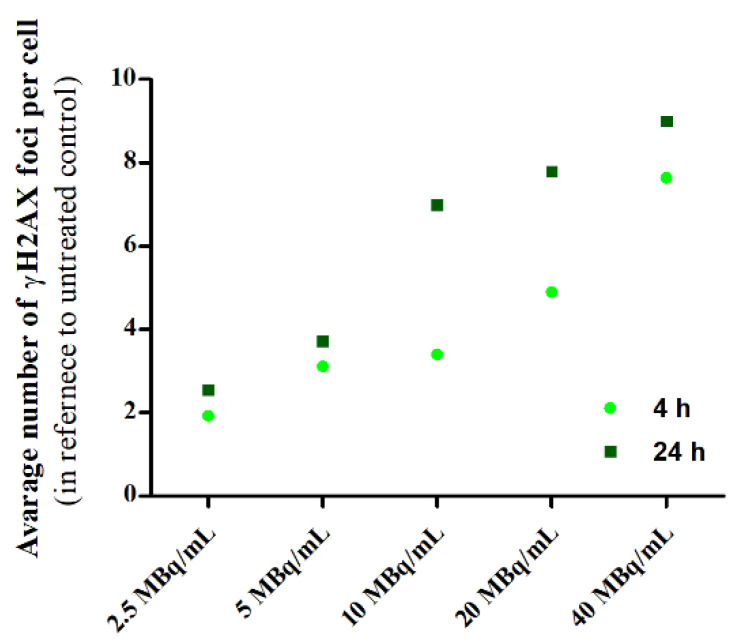
Quantification of γH2AX foci from the images shown in Figure 7.

**Table 1 ijms-25-13555-t001:** Hydrodynamic diameter and zeta potential of the synthesized conjugates.

	Au@PdNPs	Au@Pd-PEG-Panitumumab
Hydrodynamic diameter (nm)	23.78 ± 0.38	82.79 ± 0.43
Zeta potential (mV)	−35.6 ± 1.5	−25 ± 1.8

## Data Availability

The data presented in this study are available on request from the corresponding authors due to (specify the reason for the restriction).

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
