# Peer review of "Au@109Pd Core–Shell Nanoparticles Conjugated to Panitumumab for the Combined β−—Auger Electron Therapy of Triple-Negative Breast Cancer"

_ijms, 2024, doi:10.3390/ijms252413555_

Round 1
Reviewer 1 Report
Comments and Suggestions for Authors
About materials;
Q1: What is the radiochemical purity of the nanoparticles created?
Q2: Are the antibodies bound 1:1? Are they random?
Q3: How do you check the quality evaluation of the finished product?
Q4: Are the excess nanoparticles and capsules that are not the expected conjugates removed?
Q5: An overview of the developed compounds should be given as an illustration or something.
(To improve reader understanding)
Q6: Where is it bound to the radioisotope and where is it bound to the antibody? This is because it is difficult to understand what is being measured, what it shows, and what it means in the Figures that will be discussed later.
Q7: What was the procedure for forced expression of EGFR receptor (Fig2)?
Q8: It does not seem to be described in the Method. Shouldn't it be described?
Q9: If it was induced by drugs, the method should be described, and if it was created by genetic recombination, it should be indicated with the experimental license number.
Q10: The production of nuclides would also be better understood with a simple diagram.
About notation;
Q11: MDA-MB-231? or MDA-MB231? Please align the notations.
Q12: Hard to point out since the lines are not numbered, but does 8p have a different font size from the middle of the page, up to and including the 4th line?
Q13: Min or minutes should be unified.
Q14: Six-well? 6-well? I would like to see the notation of multi-wells aligned.
About figures;
Q15: The T-PMT in Fig. 4 may be intended to show the location of the cells, but it is very unclear. Shouldn't the image be slightly adjusted so that the cells are visible? At the same time, because T-PMT is difficult to see, MERGE is similarly difficult to see.
Q16: The 4h and 24hr in Fig.6 should be shown in the figure.
Q17: What is the sample size that produced the average in Fig.7?
About conclusion;
Q18: It is unclear how it will be applied in vivo. It would have been better to confirm stability using some animal-derived material, etc.
It may be understandable from someone who is familiar with this work, but if you are not very familiar with it, there are too few explanations to understand.
Author Response
Rev. 1
I am very grateful for the reviewer's comments. These were useful for improving the manuscript.
About materials;
Q1: What is the radiochemical purity of the nanoparticles created?
Answer: The radiochemical purity of Au@109PdNP-PEG-panitumumab radiobioconjugate, after centrifugation from the reaction mixture, exceeds 99%. This sentence was added to the manuscript.
Q2: Are the antibodies bound 1:1? Are they random?
Answer: On page 15 of the experimental section, there is a point titled "Determination of the number of panitumumab molecules conjugated to Au@Pd nanoparticles," which describes the procedure for assessing the number of antibody molecules per nanoparticle. As stated at the top of page 6 (after the table), 19 panitumumab molecules were conjugated to a single nanoparticle. We added “The data obtained from the DLS analysis indicate that this distribution is random”.
Q3: How do you check the quality evaluation of the finished product?
Answer: The finished product's quality was evaluated based on colloidal, chemical, radiochemical stability, and receptor affinity.
Q4: Are the excess nanoparticles and capsules that are not the expected conjugates removed?
Answer: As mentioned in the manuscript “The radioactive antibody-conjugated nanoparticles Au@109Pd-PEG-panitumumab were synthesized using the same procedure as for Au@109Pd-PEG-trastuzumab [15]” In ref.15 is mentioned “the purification of OPSS-PEG-Trastuzumab was then conducted using centrifugal concentrators Vivaspin®500 100 kDa cut-off to remove the unreacted OPSS-PEG-NHS.
Q5: An overview of the developed compounds should be given as an illustration or something.
(To improve reader understanding)
Answer: Following the reviewer’s suggestion, we have included Fig. 1 with a schematic representation of the obtained radiobioconjugate.
Q6: Where is it bound to the radioisotope and where is it bound to the antibody? This is because it is difficult to understand what is being measured, what it shows, and what it means in the Figures that will be discussed later.
Answer: In the manuscript, it is described that radionuclide 109Pd is present in metallic form on the surface of gold (Au) nanoparticles (refer to the section on Synthesis of Au@109Pd NPs). The antibody was attached to the nanoparticles in a two-step process: first, OPSS-PEG-NHS was conjugated with lysine terminal group on panitumumab, and secondly, the resulting thiol-panitumumab was conjugated to the surface of the nanoparticles through the formation of a strong Pd-thiol bond. The manuscript is a little modified.
Q7: What was the procedure for forced expression of EGFR receptor (Fig2)?
Answer: We did not force the expression of EGFR receptor. We used MDA-MB-231 human cancer cells (triple-negative breast cancer cells), which naturally overexpress EGFR receptors. Such a cells were used in another similar studies e.g. Reilly et al. Panitumumab-DOTA-111In: An Epidermal Growth Factor Receptor Targeted Theranostic for SPECT/CT Imaging and Meitner-Auger Electron Radioimmunotherapy of Triple-Negative Breast Cancer, DOI: 10.1021/acs.molpharmaceut.2c00457. Our cells were purchased from ATCC company.
Q8: It does not seem to be described in the Method. Shouldn't it be described?
Answer: The procedure wasn’t used so we didn’t describe.
Q9: If it was induced by drugs, the method should be described, and if it was created by genetic recombination, it should be indicated with the experimental license number.
It wasn’t induce by drugs or created by genetic recombination.
Q10: The production of nuclides would also be better understood with a simple diagram.
Answer: The production of 109Pd is by the nuclear reaction 108Pd(n,g)109Pd in nuclear reactor Maria. It is difficult to present this simple reaction in any other way.
About notation;
Q11: MDA-MB-231? or MDA-MB231? Please align the notations.
Answer: Done. MDA-MB-231 is correct. Thank you.
Q12: Hard to point out since the lines are not numbered, but does 8p have a different font size from the middle of the page, up to and including the 4th line?
Answer: Figure captions are in font size 10, and text is 11. I think this is allowed.
Q13: Min or minutes should be unified.
Answer: Done
Q14: Six-well? 6-well? I would like to see the notation of multi-wells aligned.
Answer: Done
About figures;
Q15: The T-PMT in Fig. 4 may be intended to show the location of the cells, but it is very unclear. Shouldn't the image be slightly adjusted so that the cells are visible? At the same time, because T-PMT is difficult to see, MERGE is similarly difficult to see.
Answer: We enhanced the contrast of the photos in visible light and merged them. Hopefully, these cells will be visible better.
Q16: The 4h and 24hr in Fig.6 should be shown in the figure.
Answer: Done. Thank you.
Q17: What is the sample size that produced the average in Fig.7?
Answer: All samples were prepared in the same way. We used MDA-MB-231 cells (2×105/well), which were cultured in 6-well plates. It is difficult to estimate the size of the sample, we can only give the information how many cells were seeded in one well.
About conclusion;
Q18: It is unclear how it will be applied in vivo
Answer: Thank you for your comment. I have added the relevant paragraph to the conclusion.
Reviewer 2 Report
Comments and Suggestions for Authors
In this article, authors discussed the potential application of 109Pd-PEG-panitumumab nanoparticle conjugates as an in-vivo 109Pd/109mAg generator for targeted therapy of TNBC. It is an interesting work, which can be accepted in IJMS after minor revision.
Here are the points:
-Authors should also investigate the cytotoxicity of this generator to healthy cells to determine the selectivity.
-There is not any detail about the chemical interactions between 109Pd-PEG-panitumumab nanoparticle and panitumumab.
-There is not also any detail about the release of panitumumab from 109Pd-PEG-panitumumab nanoparticle.
-The advantages of this therapy compared to current therapy could be discussed in detail.
Author Response
Rev.2.
I am very grateful for the reviewer's comments. These were useful for improving the manuscript.
Authors should also investigate the cytotoxicity of this generator to healthy cells to determine the selectivity.
Answer: We focused our studies exclusively on MDA-MB-231 cells. While it's true that our conjugate will also be toxic to neighboring cells due to the approximately 2-mm range of beta radiation, its accumulation and internalization in cells with EGFR receptors are significantly greater (see Fig. 3 and Fig. 5) and cytotoxicity is higher. In our previous work, we conducted similar studies using the in-vivo generator Au@109Pd/109mAg-trastuzumab, where we demonstrated that radiotoxicity was much higher in cells with Her2 receptors compared to those without.
-There is not any detail about the chemical interactions between 109Pd-PEG-panitumumab nanoparticle and panitumumab.
Answer: We do not observe any chemical interactions between 109Pd-PEG-panitumumab nanoparticles and panitumumab because 109Pd-PEG-panitumumab contains long-chain PEG chains, which act hydrophilically.
-There is not also any detail about the release of panitumumab from 109Pd-PEG-panitumumab nanoparticle.
Answer: The attachment of panitumumab to the nanoparticles occurs through a strong thiol bond. Dynamic Light Scattering (DLS) studies (see Fig. 2) indicate that there is no release of panitumumab from the nanoparticles. If there were any release of panitumumab, we would expect to see a change in the hydrodynamic radius. For clarity, we have included additional sentences in the manuscript.
-The advantages of this therapy compared to current therapy could be discussed in detail.
Answer: In conclusion, we described the advantages and disadvantages of the therapy proposed in our publication.
Round 2
Reviewer 1 Report
Comments and Suggestions for Authors
3 of 16
Section “Results and Discussion”
Line 8: “by Das et al [19]” Its font is so big. Please check.
5 of 16
In Table 1. “25” is so big font.
7 of 16
The appearance of Figure 4 should be aligned.
(font and axis writing)
8 of 16 to 9 of 16
This paragraph clearly has a different font size.
Figure 6: * How do you explain this?
Figure 7:Since the 4hr and 24hr diagrams are far apart, shouldn't the concentration be written in the 24hr diagram as well?
Figure 8
The average value is taken on the vertical axis, but is it not necessary to show the variation?
13 of 16 to 14 of 16
This paragraph clearly has a different font size.
Author Response
Editorial Board,
International Journal of Molecular Sciences
November 14, 2024
Dear Editor,
Please, find enclosed our manuscript, “Au@109Pd core-shell nanoparticle conjugated to panitumumab for the combined β- - Auger electron therapy of triple-negative breast cancer” which we would like to submit for consideration by the International Journal of Molecular Sciences, special issue “New Advances in Nanomedicine Innovation in Cancer Treatment” where Prof. Dr. Tijani Gharbis is Guest Editor.
The manuscript has been revised in accordance with the comments provided by the editor and reviewers. All remarks were corrected
Sincerely,
Aleksander Bilewicz, Professor
Head of Radiopharmaceutical Chemistry Laboratory
Centre for Radiochemistry and Nuclear Chemistry
Institute of Nuclear Chemistry and Technology
Warsaw, Poland
a.bilewicz@ichtj.waw.pl
